# Calibration and assessment of electrochemical air quality sensors by co-location with reference-grade instruments

David H. Hagan<sup>1</sup>, Gabriel Issacman-Vanwertz<sup>1,2</sup>, Jonathan P. Franklin<sup>1</sup>, Lisa M. M. Wallace<sup>3</sup>, 5 Benjamin D. Kocar<sup>1</sup>, Colette L. Heald<sup>1,4</sup>, Jesse H. Kroll<sup>1,5</sup>

<sup>1</sup>Department of Civil and Environmental Engineering, Massachusetts Institute of Technology, Cambridge, 02139, USA

<sup>2</sup>Department of Civil and Environmental Engineering, Virginia Tech, Blacksburg, 24061, USA <sup>3</sup>Air Surveillance and Analysis Section, Hawai'i State Department of Health, Hilo, 96720, USA

10 <sup>4</sup>Department of Earth, Atmospheric and Planetary Science, Massachusetts Institute of Technology, Cambridge, MA, 02139. USA

<sup>5</sup>Department of Chemical Engineering, Massachusetts Institute of Technology, Cambridge, MA, 02139, USA

15 Correspondence to: Jesse Kroll (jhkroll@mit.edu)

# Abstract

The use of low-cost air quality sensors for air pollution research has outpaced our understanding of their capabilities and limitations under real-world conditions, and there is thus a critical need for understanding and optimizing the performance of such sensors in the field. Here we describe

- the deployment, calibration, and evaluation of electrochemical sensors on the Island of Hawai'i, 20 which is an ideal test-bed for characterizing such sensors due to to its large and variable sulfur dioxide (SO<sub>2</sub>) levels and lack of other co-pollutants. Nine custom-built SO<sub>2</sub> sensors were colocated with two Hawaii Department of Health Air Quality stations over the course of five months, enabling comparison of sensor output with regulatory-grade instruments under a range
- of realistic environmental conditions. Calibration using a nonparametric algorithm (k-Nearest 25 Neighbors) was found to have excellent performance (RMSE < 7 ppb,  $r^2 > 0.997$ ) across a wide dynamic range in  $SO_2$  (<1 ppb, >2 ppm). However, since nonparametric algorithms generally cannot extrapolate to conditions beyond those outside the training set, we introduce a new hybrid linear/nonparametric algorithm, enabling accurate measurements even when pollutant levels are

higher than encountered during calibration. We find no significant change in instrument sensitivity toward  $SO_2$  after 18 weeks, and demonstrate that calibration accuracy remains high when a sensor is calibrated at one location and then moved to another. The performance of electrochemical  $SO_2$  sensors is also excellent at lower  $SO_2$  mixing ratios (

(Hodgson et al., 1999), enabling sensitive, real-time pollutant measurements. However, accurate calibration of such sensors poses a major challenge. Even setting aside the logistical difficulties associated with calibrating a large number of sensors distributed throughout a network, there are specific technical challenges that can limit the accuracy of any calibration; these include the sensitivity of sensors to environmental conditions (temperature and relative humidity) (Cross et al., 2017; Masson et al., 2015; Mead et al., 2013; Popoola et al., 2016), cross-sensitivities to other (sometimes unknown or unmeasured) atmospheric species (Lewis et al., 2015; Mueller et al., 2017; Spinelle et al., 2015), and long-term sensitivity decay (drift) associated with the evaporation of electrolyte solution (Mead et al., 2013; Smith et al., 2017).

10

5

Thus far, two primary approaches have been applied for the calibration of electrochemical (and other low-cost) air quality sensors: laboratory calibration and co-location with reference instruments. The first involves calibrating the sensor in a laboratory over a controlled and well-defined range of conditions (Castell et al., 2017; Mead et al., 2013; Piedrahita et al., 2014), as is

- 15 standard for calibration of high-fidelity atmospheric chemistry and air quality instrumentation. However, because electrochemical sensors tend to be less selective and more prone to interferences than such higher-fidelity instruments (Lewis et al., 2015), identifying and calibrating over the full range of relevant measurement conditions in the laboratory can be challenging, and the presence of additional interfering components cannot always be anticipated.
- 20 In addition, this approach requires high-quality analytical instruments and standard gas mixtures, and so is generally not an option for anyone who is not affiliated with a research institution (e.g., community organizations, citizen scientists, etc.) or is conducting research in less-well funded environments (e.g., developing countries).

The second approach for calibrating low-cost sensors is by co-location with reference instruments, typically government-run air quality (AQ) stations equipped with regulatory-grade monitors. There are multiple advantages to this approach: the reference instruments are regularly

- 5 calibrated, the reference measurement data are generally made publicly available (e.g., EPA AirNow (US EPA, 2017), OpenAQ (Hasenkopf, 2017)), and the calibrations are carried out under ambient conditions that are (at least partially) representative of the sensor measurements to be made. Indeed, the effectiveness of co-location has been demonstrated in several recent studies, with sensor outputs (voltages) and other environmental parameters (e.g., temperature)
- related to the true concentration values (from the reference instruments) via some form of regression, either from parametric models (Jiao et al., 2016; Lewis et al., 2015; Masson et al., 2015; Mueller et al., 2017; Popoola et al., 2016; Smith et al., 2017) or machine-learning/nonparametric methods (Cross et al., 2017; Spinelle et al., 2015).
- 15 While this previous work has demonstrated the effectiveness of sensor calibration by co-location, this general approach has not yet been systematically explored or optimized for realistic deployment conditions. Specifically, there has been little consideration of the performance of sensors after they are moved from the calibration location to their measurement locations. Important open topics include: ideal calibration algorithms (regression techniques), criteria for an
- 20 acceptable calibration (range of conditions sampled, length of calibration time) prior to sensor deployment, and performance of calibration algorithms when faced with conditions outside the training set. In fact, to our knowledge it has never been demonstrated whether a sensor can be calibrated at one ambient location and collect accurate data at another, which is a fundamental

requirement of any sensor deployment. Here, we attempt to address such questions by collecting an extensive co-location dataset and using it to assess various calibration algorithms. Central to this work is the development of models that are accurate, robust, repeatable, and predictive.

- All measurements in the present study are made on the Island of Hawai'i (USA); due to the ongoing eruption of Kīlauea, local levels of SO<sub>2</sub> can be extremely high (even exceeding 1 ppm) (Kroll et al., 2015), constituting serious air quality and human health concerns (Longo, 2009; Longo et al., 2010; Longo and M., 2013; Longo and Yang, 2008; Mannino et al., 1996; Tam et al., 2016). The SO<sub>2</sub> is emitted from just two point sources (Halema'uma'u and Pu'u 'Ō'ō; see
- Fig. 1) into an otherwise clean environment, leading to large spatial and temporal variability in SO<sub>2</sub> levels throughout the island. Accurate air quality measurements and estimates of human exposure to volcanic pollution ("vog") thus require a relatively dense monitoring network; in fact, the present calibration study is part of a planned island-wide AQ sensor network. Moreover, this location represents an ideal testbed for sensor characterization and validation, since air
- pollution is dominated by SO<sub>2</sub>, with no interfering gas-phase co-pollutants (H<sub>2</sub>S emissions from Kīlauea are generally quite low (Edmonds et al., 2013)), and the dynamic range in SO<sub>2</sub> can be very large (varying from <1 ppb to >1 ppm). This is in contrast to environments targeted in most other AQ sensor studies (e.g., polluted urban areas), which tend to have more pollutants, typically present at lower concentrations. This location is thus an ideal environment for the
- detailed characterization of the sensor response to a single target analyte, the focus of the present study. At the same time, because of the unique features of this environment, not all results from this work (such as accuracy of the calibration) will necessarily directly translate to other

pollutants and environments. However, the general calibration and characterization approaches described here should be suitable for use in a wide range of sensor applications.

In this study, we install a set of low-cost, autonomous SO<sub>2</sub> sensor nodes at AQ stations on the

- 5 island for a period of five months. This provides a large dataset for testing, validating, and optimizing this in-field co-location approach to calibration. We evaluate a number of sensor calibration algorithms (both parametric and nonparametric), with a particular focus on the temperature dependence of the baseline. Further, we investigate the performance of the calibrations given practical constraints (e.g., the possibility that measurement conditions may be
- 10 different from those of the calibration period), and examine how sensitivity changes over a period of several months.

# 2 Experimental Techniques and Design

## 2.1 Sensor Node Design

Measurements were made using a custom sensor node for continuous, real-time monitoring of ambient SO<sub>2</sub> and environmental variables (temperature, relative humidity) at a fixed-site location. Each node is powered by a small solar panel and is internet-connected via a 3G cellular module to allow bi-directional communication between a server and the sensor node. The nodes are weatherproof (housed in a UL-certified weather-proof enclosure) and low-power (~1W), with
 a total component cost of ~\$400. Major components of the design are shown in Fig. 2.

SO<sub>2</sub> is measured using an Alphasense SO2-B4 electrochemical sensor (purchased December 2016, opened January 2017) in conjunction with the Alphasense potentiostat circuitry. This 4-

electrode sensor includes a working electrode (WE), at which the electrochemical reaction (oxidation of SO<sub>2</sub>) takes place, as well as an auxiliary electrode (AE), which is isolated from the gas phase, but responds to changes in the signal associated with changing environmental variables. In particular, it has been shown that the AE response to changes in ambient

- temperature and relative humidity is non-linear (Cross et al., 2017; Lewis et al., 2015; Masson et al., 2015; Mead et al., 2013) and depends on not only these parameters but also their derivatives (Masson et al., 2015; Pang et al., 2017). The SO<sub>2</sub> sensor and adjacent relative humidity and temperature (RHT) sensor (HIH6130, Honeywell) are embedded in a 3D-printed flow chamber, with a small direct current (DC) fan used to pull air perpendicular to the surface of the sensors.
- This design is improved from an earlier prototype that used a passive external sensor, which was susceptible to large temperature variations caused by direct irradiation by sunlight, and may have exhibited poorer sensitivity (Masson et al., 2015). The inlet and outlet of the recent design are protected from the elements by 3D-printed awnings that are epoxied in place.
- The analog signals are sampled at 20 Hz using a 16-bit Analog-to-Digital (ADC) converter (Texas Instruments ADS1115), before being averaged and saved locally as a 1-Hz measurement on a micro-SD card. The 1-Hz measurements are then averaged over a user-defined interval and transmitted to a remote server where data is stored in a MySQL database and visualized in realtime. Flags were set to mark the first four hours after a node was turned on to indicate a sensor
- warm-up period (Roberts et al., 2012; Smith et al., 2017). In addition, flags are set whenever the ADC or RHT sensor reported a failure. Throughout this deployment, data are transmitted to the server at a 1-minute interval. The node is operated using a 3G-enabled, ARM-based microcontroller (Particle Electron), allowing for two-way communication between the node and

5

the server. Each node is powered continuously using a 9W solar panel (Voltaic Systems) with a 4000 mAh battery (Voltaic Systems V15) serving as the power supply when the solar panels are not supplying enough power. In areas with less sunlight, two 6W panels in parallel are used rather than a single 9W panel. At full charge the battery can supply continuous backup power for 20 hours, allowing the nodes to run overnight without loss of power.

#### 2.2 Co-Location Details

### 2.2.1 Site Description and Reference Data

Sensor nodes were first deployed on the island of Hawai'i beginning January 15th, 2017 and most

- 10 are still active as of August 2017. The Hawaii Department of Health (DOH) operates six AQ monitoring stations that continuously monitor SO<sub>2</sub> and supporting meteorological variables including wind speed, wind direction, relative humidity, and temperature (Hawaii Department of Health, 2017). Continuous SO<sub>2</sub> measurements are made by a pulsed-fluorescence analyzer (Thermo Scientific 43i), which provide data as 1-minute averages, and are calibrated at least
- once every two weeks. The data are continuous except during periods of calibration, which are excluded from the dataset. The AQ stations are spread across the island; the two primary sites used in this work are Pahala and Hilo (see Fig. 1). Pahala (pop. ~1,300; location: 19°12'9" N, 155°28'38" W) is located 37 km southwest of the main volcanic vent (Halema'uma'u), and so is subjected to the volcanic plume when the trade winds (the prevailing winds, from the northeast)
- are dominant. The mean 1-hour SO<sub>2</sub> level is 39 ppb, though levels can exceed 1 ppm during direct plume hits (typically in the morning, when the boundary layer is low) (Kroll et al., 2015). Hilo (pop. ~43,300; location: 19°42'20" N, 155°5'9" W) is located 50km northeast of the

volcanic vent and is characterized by much lower SO<sub>2</sub> values, with a mean 1-hour level of 6 ppb and a yearly maximum of 500 ppb (during southwesterly "Kona winds").

### 2.2.2 Co-Location of Nodes

- Nine sensor nodes were installed at the Pahala AQ station for no less than 48 hours each over a four-day period (January 15<sup>th</sup> January 19<sup>th</sup>, 2017) for initial calibration. (Two additional nodes lost power for some fraction of this calibration period, and thus are not included in this study.) At the end of this calibration period, two nodes were re-located to the Hilo AQ station (January 23<sup>rd</sup>, 2017 ongoing as of August 2017), and three nodes remained at Pahala (still operating as of
- August 2017). The remaining four nodes were distributed to elementary and middle schools across the island; due to the lack of co-location data, measurements taken at the schools will not be discussed here. All co-located nodes were mounted on the roof of the air quality monitoring station, within 2 meters of the reference instrument's inlet. In this work, we focus on the data collection period of January 15<sup>th</sup>- May 22<sup>nd</sup>, 2017. Power loss due to lack of sufficient sunlight
- impacted several nodes (mostly during early morning periods), though the two nodes located at Hilo and one node located at Pahala suffered no power loss. Beginning April 25<sup>th</sup>, the RHT sensor on one of the Pahala nodes (SO2-02) began to behave erratically for hours at a time, making it difficult to assess the data beyond that date.

## 20 2.3 Data Analysis

#### 2.3.1 Data Preparation

A time delay between the sensor data and AQ station reference data caused by differences in clock times and inlet residence times was corrected by determining the maximum cross-

5

correlation (typically ~3 min) between the two time-series (Knapp and Carter, 1976). Measurements marked by flags (indicating calibration of the reference instrument, sensor warmup time after power-on, etc.) were removed in both data streams prior to removing all sensor data for which no reference data was available. This process led to the exclusion of less than 1% of all sensor data collected.

2.3.2 Sensor Calibration Appro