# Peer review of "Calibration and assessment of electrochemical air quality sensors by co-location with reference-grade instruments"

_Atmospheric Measurement Techniques, 2017_

## Referee Comment (RC1) · Anonymous Referee #1 · 2 Oct 2017

This work presents a detailed analysis of the performance one type of electrochemical air quality sensor for SO2 detection. The authors use data from multiple sensors deployed for approximately 21 weeks and compare with co-located reference-grade SO2 instruments on the island of Hawaii. The performance of multiple regression methods to calibrate the electrochemical sensors and correct for known temperature responses are evaluated. The availability and interest in low cost sensor technologies over recent years means comprehensive evaluations of their performance and possible sampling methodologies such as this are essential. As acknowledged by the authors, the choice of Hawaii as a sampling location provides the best possible scenario for sensor performance, due to the large dynamic range of SO2 mixing ratios experienced and the

lack of any significant interfering co-pollutants. How well the sensors and analytical methods presented in this work would perform in other environments (e.g. urban) is therefore still questionable. The manuscript is clear and well written, and presents one of the most comprehensive assessments of low cost sensor performance to date. I recommend publication after the following minor comments have been addressed.

Minor comments

1. In section 3.5 and Fig. 7 it would be useful to compare with the performance of one of the sensors that remained at the Pahala site that was trained only using data from the same 2 days. Adding this data to Fig.7 would help demonstrate the decreased performance of the regression used for calibration due to environmental parameters compared to the change in training data fraction (changing from 70% in earlier Figs. To <2% in Fig. 7).

Typographical errors

1. Unsure if this was just a problem with my version but Hawaii often spelt Hawai'i

2. Page 16 line 7: Figure ?

3. Page 19 line 12: Figure 7 should read Figure 8

---

## Referee Comment (RC2) · Anonymous Referee #2 · 3 Oct 2017

This is a well written manuscript that provides a potentially useful mathematical framework and method for extrapolating and interpreting lower cost sensor data against reference grade instrumentation. The work involves a large dataset of both experimental and reference data, and a thorough review of some simple machine learning routines to improve sensor predictions. The manuscript itself is particularly clear in its writing, and is easy to follow. I would recommend publication if the authors could address the following comments:

Minor Comments: There was an inconsistent use in the spelling of Hawaii (or Hawai'i) throughout the manuscript.

Major Comments:

Temperature dependence still seems to be an issue and warrants further explanation. For example, in Fig 7, it appears that low temps bias predicted concentration high, and high temps bias the predicted concentration low. This isn't always true in their data, but it would be useful if the authors would describe why such a consistent bifurcation persists.

The conclusion that this regression algorithm 'can be applied to any other sensor system' (page 22, line 16) seems to be an over reach and is without supporting data. While one may certainly elect to apply any algorithm to any dataset, whether it is useful or not remains to be seen. This is particularly true if one were sampling in strictly ambient levels for gases, where the wide dynamic range observed in this study would not necessarily exist.

While the study location is certainly convenient for observing a wide, dynamic range of SO2, this range is very unlikely to be observed in many other places across the world. The authors correctly note this as a limitation of the study, but the range of largest uncertainty is precisely where typical ambient concentrations of SO2 live. Describing (Fig 8) as 'lower levels of SO2' as less than 50ppb seems misleading; SO2 concentration in the US and across most of the EU is less than 20 ppb and trending lower. Further, the data in Fig 8 appear much tighter in the 20-50ppb range, and may be strongly biasing the regression.
* * *

---

## Author Comment (AC1) · 29 Nov 2017

We would like to thank both reviewers for their helpful feedback. We have addressed all their comments in the revised version of the manuscript, as described below; the revisions mostly involve clarifications to the text and changes to the way the sensor data are presented in Figures 7 and 8. In addition, the reviewers' comments have motivated us to make two minor changes to the calibration approach taken, specifically the details of the hybrid algorithm, and the use of relative humidity as a calibration parameter at low SO2 concentrations. Before, the hybrid algorithm used only  $V_{WE}$  as a proxy for SO2 concentration, but this was unable to account for the temperature effect at low concentrations, leading to a small portion of the data points being incorrectly classified as "high SO2" points. To combat this, we have chosen to use a classification algorithm instead as detailed in Section 3.4 (Page 17 lines 5-20) resulting improved classification of inputs to the correct algorithm. Such changes are minor, and have no effect on any of the conclusions of the work.

Below we respond to the reviewers' specific comments in detail. Reviewer comments are in black text while **author comments are in bold blue text**.

**Response to Reviewer 1**

This work presents a detailed analysis of the performance [of] one type of electrochemical air quality sensor for SO2 detection. The authors use data from multiple sensors deployed for approximately 21 weeks and compare with co-located reference-grade SO2 instruments on the island of Hawaii. The performance of multiple regression methods to calibrate the electrochemical sensors and correct for known temperature responses are evaluated. The availability and interest in low cost sensor technologies over recent years means comprehensive evaluations of their performance and possible sampling methodologies such as this are essential. As acknowledged by the authors, the choice of Hawaii as a sampling location provides the best possible scenario for sensor performance, sur to the large dynamic range of SO2 mixing ratios experienced and the lack of any significant co-pollutants. How well the sensors and analytical methods presented in this work would perform in other environments (e.g. urban) is therefore still questionable. The manuscript is clear and well written, and presents one of the most comprehensive assessments of low-cost performance to date. I recommend publication after the following minor comments have been addressed.

 In section 3.5 and Fig. 7 it would be useful to compare with the performance of one of the sensors that remained at the Pahala site that was trained only using the data from the same 2 days. Adding this data to Fig. 7 would help demonstrate the decreased performance of the regression used for calibration due to environmental parameters compared to the change in training data fraction (changing from 70% in earlier Figs. To <2% in Fig. 7).

This is an excellent suggestion. We have updated the way we operate the hybrid algorithm (detailed in Section 3.4 of the updated manuscript) which has slightly changed the output for Fig. 7. Along with updating the figures, we have added a comparison plot for sensor S-02 (which remained at Pahala) which can be found in the SI. We have also added more text to section 3.5 to more clearly convey the importance of ensuring the training data is similar in structure to the validation data.

2. Unsure if this was just a problem with my version but Hawaii often spelt Hawai'i

The correct spelling of the island is "Hawai'i", including the 'okina (a letter in the Hawaiian alphabet that looks like an open single-quotation mark), whereas the official spelling of the state is Hawaii, with no 'okina. Thus, both "the Island of Hawai'i" and "Hawaii Department of Health" are correct.

3. Page 16 line 7: Figure ?

This has been corrected to Figure 6.

4. Page 19 line 12: Figure 7 should read Figure 8

Figure 7 has been changed to Figure 8 on line 9 of Page 19.

**Response to Reviewer 2**

This is a well written manuscript that provides a potentially useful mathematical framework and method for extrapolating and interpreting lower cost sensor data against reference grade instrumentation. The work involves a large dataset of both experimental and reference data, and a thorough review of some simple machine learning routines to improve sensor predictions. The manuscript itself is particularly clear in its writing, and is easy to follow. I would recommend publication if the authors could address the following comments:

1. There was an inconsistent use in the spelling of Hawaii (or Hawai'i) throughout the manuscript

As discussed in comment #2 by Reviewer 1, "Hawai'i" refers to the island whereas "Hawaii" refers to the state.

2. Temperature dependence still seems to be an issue and warrants further explanation. For example, in Fig 7, it appears that low temps bias predicted concentration high, and high temps bias the predicted concentration low. This isn't always true in their data, but it would be useful if the authors would describe why such a consistent bifurcation persists.

We believe this bifurcation exists because the temperature and  $SO_2$  range we trained in was different than the validation set (see subplots of Figure 7 and updated description in section 3.5 (Page 18, lines 8-25)). Also, compared to Fig. 5, the ratio of training to validation data is much lower. As a result of more data being included in the training data, there is broader coverage of the entire temperature/SO2 space. Additional discussion of topic has been added (Page 18, lines 8-25).

3. The conclusion that this regression algorithm 'can be applied to other sensor system' (page 22, line 16) seems to be an over reach and is without supporting data. While one may certainly elect to apply any algorithm to any dataset, whether it is useful or not remains to be seen. This is particularly true if one were sampling in strictly ambient levels for gases, where the wide dynamic range observed in this study would not necessarily exist.

We agree with the reviewer's comments; we have softened the language of the text to indicate that we are not trying to claim this approach will work for other sensor systems, but rather the hybrid algorithm approach could be tried to alleviate issues associated with common nonparametric ML techniques to enable extrapolation outside the training dataset. The revised text (Page 22/23, lines 20-5) is as follows:

"However, the general approaches discussed here – the use of a hybrid linear/nonparametric regression algorithm, the examination of calibrations by limiting the environmental conditions of the training set, and the testing of sensors and algorithms by calibration at one reference site and validation at another – could be applied to other sensor systems as well; sensor characterization in these other conditions is an important area of future research."

We highlight this uncertainty in other parts of the text as well (e.g., the last sentence of the abstract (Page 2, lines 5-10) and the second-to-last paragraph of the introduction (Page 4, line 16 – Page 5, line 4)).

4. While the study location is certainly convenient for observing a wide, dynamic range of SO2, this range is very unlikely to be observed in many other places across the world. The authors correctly note this as a limitation of the study, but the range of largest uncertainty is precisely where typical ambient concentrations of SO2 live. Describing (Fig 8) as 'lower levels of SO2' as less than 50 ppb seems misleading; SO2 concentration in the US and across most of the EU is less than 20 ppb and trending lower. Further, the data in Fig 8 appear much tighter in the 20-50 ppb range, and may be strongly biasing the regression.

We understand the concern of the reviewer, and have changed the range of "lower SO2" from 0-50 ppb to 0-25 ppb. While this may still be in the upper percentile of observed SO2 in the US and parts of Europe, much of the world's population lives in locations (India, China, etc) where SO2 regularly exceeds these values.

We have re-run our evaluation using SO2 concentrations between 0-25 ppb and have updated the manuscript accordingly. In addition, when evaluating at these lower levels of SO2, the influence of RH is non-negligible, and thus has been added as a parameter in the k-nearest neighbors regression algorithm as discussed in the updated manuscript [Page 19, lines 17-22].